# OpenReview forum: "Emergence of Biased Consensus in Multi-Agent LLM Debates"
_ICML.cc/2026/Conference — ICML 2026 regular_

### Official Review · Reviewer_eLa4 · 2026-02-17

**Soundness:** 3
**Presentation:** 2
**Significance:** 3
**Originality:** 3
**Overall Recommendation:** 5
**Confidence:** 3

**Summary:**

The paper investigates the conditions for a multi-agent debate to arrive a biased consensus by modeling the interactions between the LLMs as a physical spin model and mean-field theory.
The authors utilize this model to predict how a consensus evolves over debate stages based on parameters defining the temperature, how easily the agent is influenced (in effects such as group-conformity or sycophancy), internal bias and visibility of other agents.
The authors measure the consensus over several experiments including synthetic data and real world examples and show that their model accurately predicts the change of consensus over time.

**Compliance With Llm Reviewing Policy:**

Affirmed.

**Final Justification:**

My main issues with the papers was its presentation of the theory, which is not trivial to people with a background in physics. The authors agreed and also clarified other misunderstandings. Furthermore, they presented their plan to revise the paper to be more pedagogic. I trust the authors to follow this plan and consequently decided to increase my score from 4 to 5.

**Key Questions For Authors:**

1. The authors start explaining the model in line 172 directly with the Hamiltonian "Agents interact via a
local energy (cost) function". This is quite abrupt, and does not explain how the agents interact with each other through this energy function. Can the authors explain the interaction before, for a smoother transition into the theory?
2. The kernel function $\phi$ is defined differently for the binary and multi-choice models, where in the binary case it returns the values $\pm 1$ and in the multi-choice it returns ${0,1}$. Can the authors expand on this design choice? Can they also expand the choice and meaning for the negative sign of the Hamiltonian  (eq. 1)?
3. Line 184 introduces the preferred option $\sigma^{*}$ without a definition. What is the definition of the preferred option? For whom is it preferred? is that supposed to be $\sigma_{i}(0)$ for some $i$?
4. Can the authors clarify eq. 3? Is is taken directly from prior work, i.e. Goldenfeld 2018? Can the authors please also clarify the idea of critical point when $\beta J=1$? Why does it not depend on $h$?
5. In lines 223-228, the authors rewrite the "external field" as the sum of $h^{\text{neutral}}$ and $h^{\text{bias}}$, but they are not clearly defined. What are concrete examples for each of them? Why is this decomposition necessary?
6. Lines 268-269 introduce the idea of crossover regime, but this term is not defined in the paper. Could the authors define it, and perhaps provide examples?
7. Line 223 (right column) claims that $\rho\left(N-1\right)\approx\rho N$ is well-approximated. However since $N$ is a small integer, the difference can be more than 10%. Also $N$ is larger than the other parameters in the formulation. Can they authors explain the meaning of "well-approximated" and its significance?
8. In Figure 2, (a-b) show $\beta$ as the x-axis, but in (c-e) the x-axis is the $T$ in reverse. I assume this is done because of the inverse relation between $\beta$ and $T$. Is that the case? For consistency, would it be helpful to use the same axis for all the subfigures?
9. In the caption of Figure 4, the confidence visibility is marked with $\lambda\downarrow$, but the text (lines 355-356, right column) says "increasing $\lambda$". Which one is correct, should $\lambda$ increase or decrease?
10. Line 364 defines $\left|m\left(R\right)\right|=\left|\sum_{j=1}^{N}s\left(\sigma_{j}\right)\right|$. What are $R$ and $s$? Why is it a function of $R$?
11. Lines 378-383 state "As sampling temperature $T$ increases [...] the debate rapidly converges to a collective norm aligned with that bias". But according to the theory and other experiments, increasing the temperature should prevent a biased consensus. Is this a contradiction, a typo, or a misunderstanding?
12. The paper suggests to increase the temperature to avoid bias. What are the downsides and limitations of increasing the temperature?

**Limitations:**

Yes.

**Strengths And Weaknesses:**

## Strengths
1. The paper presents multiple experiments that convincingly support its claims.
3. The authors are upfront about the limitations of their analysis and experiments.
4. The paper tackles a realistic problem and discusses practical solutions.
5. The paper elegantly utilizes a physical model to analyze the shift in consensus in a multi-agent debate format.

## Weaknesses
1. The presentation of the model is not clear: missing some background and some terms are not introduced appropriately.
2. There are some inconsistencies in the text and figures (please refer to the questions below for specifics).
3. The paper lacks a discussion about bias and consensus (please see relevant questions below).

To summarize, I believe this is a decent paper, with an interesting model and experiments verifying the theory. However, I refrained from a higher score due to the theory not being detailed or clear enough.
For example, the definition of $m(t)$, the "mean opinion" of the collective, is barely defined in line 213, even though it is a crucial statistic for the rest of the paper and experiments and could use a more detailed discussion (it is also used in line 212 before it is defined).
In my questions, I list other details and clarification I believe are missing.
If the authors will address these issues and incorporate them in the next revision, I will be open to updating my scores.

---

> ### Author Rebuttal · Authors · 2026-03-30
>
> Thank you for the careful and constructive reading. We appreciate your positive assessment of the paper’s *elegance*, *practical relevance*, and *experimental support*. Below, we respond to each comment in turn.
>
> > **Question 1**
>
> We motivate our model using the standard analogy to spin systems, where local interactions give rise to macroscopic order. In our setting, local social influence among LLM agents produces collective behavioral patterns such as bias amplification. Under this analogy, the Hamiltonian acts as a cost function that aggregates pairwise alignment incentives and individual predispositions, thereby formalizing the tension between social conformity and intrinsic opinion.
>
> > **Question 2**
>
> Our formulation follows the standard distinction between Ising and Potts models. In the binary case, disagreement contributes with an opposite sign, inducing explicitly antagonistic interactions. In the multi-choice case, the interaction term rewards exact agreement without introducing a signed notion of opposition between categories, reflecting categorical rather than ordinal structure. The negative sign in Eq. (1) follows the usual convention that aligned configurations correspond to lower energy.
>
> > **Question 3**
>
> The preferred option $\sigma^\ast$ is defined by the external field and represents a prior bias independent of interactions. It is conceptually distinct from the initial condition $\sigma_i(0)$, which specifies the agents’ starting states. Rather, $\sigma^\ast$ encodes the direction of the predisposition imposed by the field. In the homogeneous setting, this preference is shared across agents, hence no index is required.
>
> > **Question 4**
>
> Eq. (3) corresponds to the standard mean-field recursion for the binary Ising model. The aggregate state $m(t)$ feeds back into $m(t+1)$ via the interaction term $Jm(t)$, yielding a self-consistent dynamical system. Linearizing near $m=0$ gives $m(t+1)\approx \beta J m(t)+\beta h$. At $h=0$, the neutral fixed point is stable when $\beta J<1$ and unstable when $\beta J>1$, with $\beta J=1$ marking the bifurcation threshold. A nonzero field $h$ selects the direction of alignment but does not alter this linear stability condition.
>
> > **Question 5**
>
> We decompose the external field into $h_i^{\mathrm{neutral}}$ (correctness-aligned) and $h_i^{\mathrm{bias}}$ (systematic bias). The former represents a preference for ground-truth or logically consistent answers (e.g., the correct solution in a reasoning task), while the latter captures intrinsic model tendencies unrelated to correctness (e.g., stylistic or positional biases). In our experiments, we intentionally select tasks without a unique ground truth to isolate bias dynamics, effectively setting $h_i^{\mathrm{neutral}}=0$ and focusing on the amplification of $h_i^{\mathrm{bias}}$.
>
> > **Question 6**
>
> Following Privman & Fisher (1984), we use “crossover” to denote the finite parameter range near the transition where behavior changes smoothly due to finite system size. In our setting, the control parameter is $\lambda \bar z / T$, and the observable is $|m(R)|$, the magnitude of the final collective opinion. Around $\lambda \bar z / T \approx 1$, the system transitions rapidly but continuously from weak alignment to strong consensus. Unlike the infinite-size limit, this transition is spread over a finite interval, which we refer to as the crossover regime.
>
> > **Question 7**
>
> We agree that $\bar z \approx \rho N$ is too crude for small $N$. We will replace it with the more accurate finite-$N$ expression $\bar z \approx \rho (N-1)$.
>
> > **Question 8**
>
> The theoretical plots are parameterized by inverse noise (i.e., effective inverse temperature), whereas empirical plots are shown directly as a function of $T$, leading to an apparent reversal of the x-axis.
>
> > **Question 9**
>
> The correct interpretation is increasing confidence visibility $(\lambda_i \uparrow)$, not decreasing.
>
> > **Question 10**
>
> We define $m(R)=\frac{1}{N}\sum_{j=1}^N \sigma_j(R)$ as the average opinion at the final round $R$. Here, $R$ denotes the total number of debate rounds, and $\sigma_j(R)$ is agent $j$’s state at that round. Thus, $m(R)$ depends on $R$ through the iterative update dynamics.
>
> > **Question 11**
>
> This was a typo: decreasing $T$, not increasing $T$, promotes rapid lock-in to a biased consensus.
>
> > **Question 12**
>
> We agree that increasing $T$ is not universally beneficial. While higher temperature mitigates premature lock-in by injecting stochasticity, excessively high $T$ leads to near-random responses and prevents stable convergence. Although a full analysis of the high-temperature regime is beyond the scope of this work, we will explicitly note this tradeoff as a limitation.
>
> ---
>
> We appreciate your openness to updating your score. We hope our responses have addressed your concerns, and we would greatly appreciate your reconsideration of your score.

---

> > ### Author Rebuttal · Reviewer_eLa4 · 2026-04-01
> >
> > The authors answered all my questions, but did not resolve some of my concerns.
> > Specifically, I am looking for how they will revise the paper to better introduce their ideas.
> >
> > For example, for points 1, 2 and 4, the authors based their answer on the standards of the Ising and Potts models, which are not popular or studied much in the ML community. For this reason, some concepts, such as the Hamiltonian and mean-field, should be more carefully introduced and explained.
> > For point 3, I am still unclear about the meaning of the preferred option and how its value is determined.
> > I also believe that the paper should better explain points 5, 6 and 10.
> >
> > In their rebuttal, the authors explicitly state they will include the trade off limitation (12) in the revision, but they do not state it in any other question. At the moment, it is unclear what the revised paper will include, and I am not increasing the score.

---

> > > ### Author Response · Authors · 2026-04-03
> > >
> > > Thank you again for the careful and constructive reading. Your comments are very helpful, and we believe they will substantially improve the paper. We fully agree with your central concern: the current presentation does not sufficiently introduce the ideas in a way that is accessible to ML readers.
> > >
> > > Due to space constraints in the rebuttal, we were not able to explicitly describe all planned revisions. We clarify here that **all of your points will be incorporated into the revision**.
> > >
> > > ---
> > >
> > > ### Restructuring the introduction of the model and analogy (Points 1, 2, 4)
> > >
> > > We agree that the current presentation introduces the Hamiltonian and spin model formalism too abruptly (Point 1).
> > > To address this, we will **restructure Section 4** so that the LLM-based intuition is introduced before any formalism.
> > >
> > > Specifically, we will first explain how LLM agents influence each other through interaction, and then explicitly map the components of the model:
> > >
> > > - agents $\rightarrow$ LLM instances
> > > - states $\rightarrow$ discrete answers
> > > - interactions $\rightarrow$ exposure to other LLM agents' responses
> > > - energy (Hamiltonian) $\rightarrow$ a cost capturing disagreement and bias
> > >
> > > We will then introduce Eq. (1) as a cost function balancing (i) social conformity and (ii) intrinsic preference (Point 2), and explain mean-field dynamics as each agent responding to the current group average (Point 4), supported by step-by-step intuition and a schematic illustration.
> > >
> > > This restructuring will ensure that the model is understandable from an LLM perspective without requiring prior knowledge of Ising/Potts models.
> > >
> > > ---
> > >
> > > ### Clarifying the preferred option (Point 3)
> > >
> > > We will explicitly introduce the preferred option before its first use.
> > > Intuitively, the preferred option $\sigma^\ast$ is the answer that an LLM is intrinsically more likely to produce due to its internal biases (e.g., training data or positional effects), even before observing other agents. In contrast, the initial condition $\sigma_i(0)$ is a particular set of sampled outputs at the first round, which may deviate from this preference due to stochastic decoding.
> > >
> > > ---
> > >
> > > ### Clarifying key concepts (Points 5, 6, 10)
> > >
> > > We will clarify these concepts with explicit definitions and concrete LLM-based interpretations.
> > > Specifically, we will define the external field (Point 5) as a bias over answer choices. Intuitively, it represents an intrinsic preference of an LLM toward certain answers (e.g., due to training data or positional effects), independent of interactions.
> > > We will introduce the crossover regime (Point 6) as the finite-$N$ analogue of a phase transition, where the group no longer jumps abruptly to consensus, but instead becomes gradually more consistent as temperature changes; and define the final collective opinion $m(R)$ (Point 10) as the average of LLM agents' answers at the final round, emerging from iterative updates where each agent responds to the current group average.
> > >
> > > The definitions and intuitions will be introduced upon first use.
> > >
> > > ---
> > >
> > > We will also incorporate all remaining corrections: replacing approximations with accurate finite-$N$ expressions (Point 7), clarifying figure parameterizations (Point 8), fixing typos (Point 9, 11), and explicitly discussing the trade-off of temperature as a limitation (Point 12).
> > >
> > > These revisions are not minor clarifications but a restructuring of the presentation that makes the LLM-based intuition primary and allows the model to be understood without prior background in statistical physics.
> > > We believe these changes address your concerns and significantly improve the clarity of the paper. We would greatly appreciate your reconsideration of the score, and we would be happy to clarify any remaining questions.

---

### Official Review · Reviewer_bYP4 · 2026-03-07

**Soundness:** 3
**Presentation:** 4
**Significance:** 2
**Originality:** 1
**Overall Recommendation:** 4
**Confidence:** 4

**Summary:**

This paper studies the emergence of biased consensus in multi-agent LLM debates. The core idea is to model debate dynamics through a social-dynamics / Ising-style lens, where collective behavior is shaped by conformity, local bias, and sampling noise. The paper combines this analytical framing with controlled experiments on synthetic binary-choice and implicit-bias tasks, and then evaluates whether the same qualitative picture extends to two more realistic settings: investment recommendation and LLM-as-a-judge. A central empirical claim is that low temperature can drive small individual biases into a strong collective norm, while heterogeneity and other design choices can smooth or suppress this effect.

**Compliance With Llm Reviewing Policy:**

Affirmed.

**Final Justification:**

The authors addressed most of my concerns.

**Key Questions For Authors:**

1. Why is U.S./technology concentration the right definition of bias for the investment task? Can the authors show that the main conclusions are robust to alternative definitions, such as concentration, diversification, benchmark deviation, or risk-adjusted performance measures?

2. What is the precise novelty claim of the paper? Should the main contribution be understood as a new theory, or primarily as an application/interpretation of an existing modeling framework to LLM debates?

3. How should a practitioner use this framework in deployment? If the theory is mainly diagnostic, what concrete decision would it change in the design of a real multi-agent LLM system?

4. Do the authors believe bias and performance should be expected to be inversely related in general, or only in the specific tasks and metrics considered here?

I am am willing to reconsider my score if the authors address my concerns and questions in a satisfactory manner.

**Limitations:**

Yes.

**Strengths And Weaknesses:**

### Strengths

1. **Clear and well-written paper.**
   The presentation is one of the strongest aspects of the submission. The motivation is easy to understand, the theoretical setup is introduced in an intuitive way, and the paper does a good job walking the reader from the motivating examples to the social-dynamics formulation and then to the experiments. Even where I have reservations about the ultimate contribution, I found the paper readable and conceptually clear.

2. **Appealing theory/empirics alignment.**
   The Ising/social-dynamics framing is a natural and interesting way to think about consensus formation in multi-agent debate. In particular, the phase-diagram style presentation and the qualitative agreement between the finite-$N$ theory and the observed empirical behavior are compelling. The paper does a nice job showing that low temperature and sufficient local bias can lead to rapid lock-in, and that heterogeneity tends to smooth this effect.

3. **More than a purely synthetic study.**
   I appreciate that the paper does not stop at toy tasks. In addition to controlled synthetic experiments, it also evaluates the proposed perspective on investment recommendation and LLM-as-a-judge, which makes the work more interesting than a purely stylized demonstration.

### Weaknesses / Major Concerns

1. **The definition of "bias" is not well justified in the realistic tasks.**
   My main concern is construct validity. In the investment task, the paper defines bias through U.S./technology concentration. I do not find this definition sufficiently justified. A portfolio tilted toward U.S. or technology assets could reflect many things other than an undesirable bias, including benchmark composition, growth expectations, momentum, or legitimate market views. Because the paper's realistic-task conclusions depend heavily on this metric choice, it is hard to know whether the paper is actually measuring "bias" in a meaningful sense, as opposed to a somewhat arbitrary portfolio characteristic. This concern is less severe for self-preference in LLM-as-a-judge, but for investment recommendation it feels especially under-motivated.

2. **Novelty seems limited.**
   The paper is interesting, but I am not fully convinced by the novelty. The submission itself positions the approach in relation to existing social-dynamics and collective-behavior perspectives, and as a result the contribution reads more as a transfer and adaptation of an existing modeling lens to multi-agent LLM debates than as a fundamentally new theoretical development. That can still be valuable, especially when supported by experiments, but it limits how original the core contribution feels.

3. **Limited actionability.**
   I also do not find the paper especially actionable. The main interventions suggested by the theory are things like changing temperature, sparsity, or heterogeneity, and the paper does show that heterogeneous ensembles can reduce the measured bias and sometimes improve performance on the tasks studied here. However, this still feels more diagnostic than prescriptive. The paper does not establish that reducing the chosen bias metric will generally improve performance, nor does it provide a robust recipe for practitioners beyond broad qualitative guidance. In other words, the framework helps explain when biased consensus may arise, but it is less clear how much it helps someone actually design reliable multi-agent systems in a principled way across tasks.

---

> ### Author Rebuttal · Authors · 2026-03-30
>
> Thank you for your insightful suggestions. We are encouraged by your remarks that the paper is *“conceptually clear,”* *the theory/empirics alignment is “compelling,”* and the framing is *“natural and interesting.”* We address your points below.
>
> ---
>
> ### Question (and Weakness) 1
> > Why is U.S./technology concentration the right definition of bias?
>
> Thank you for pointing this out. We emphasize that U.S./technology concentration is used as a motivating example, rather than a definitive notion of bias.
> In our framework, bias is defined more generally as a systematic preference encoded in the external field $h_i^{\mathrm{bias}}(\sigma_i)$, independent of task-aligned correctness. Empirical metrics (e.g., portfolio concentration) serve as task-specific proxies for observing this underlying phenomenon. Our claim does not depend on any particular metric, but on the dynamics by which small initial asymmetries are amplified into stable consensus.
> To complement our results on the investment recommendation task, we conducted additional experiments on the OpinionQA dataset (please see **Tables 1 & 2** in the response to Reviewer 2UJw), where bias is defined independently as deviation from human demographic distributions. We observe the same pattern: bias is amplified at low temperature and suppressed at higher temperature. This consistency suggests that the phenomenon is not tied to a specific metric, but reflects a general property of multi-agent LLM systems.
> We will further extend our evaluation to additional high-stakes domains, including hiring and legal decision-making, in the revision.
>
>
> ---
>
>
> ### Question (and Weakness) 2
> > What is the precise novelty claim of the paper?
>
> Thank you for raising this important point. Our contribution goes beyond applying existing social-dynamics models in three aspects:
> 1. We identify *bias amplification as a phase-transition-like phenomenon* in multi-agent LLM interactions, where even small initial asymmetries are non-linearly amplified into stable collective norms. While related concepts exist in social dynamics, this phenomenon has not been characterized or empirically validated in LLM-based systems.
> 2. We treat collective behavior as an emergent *phenomenon* and introduce a physics-inspired methodology that combines *theory-driven hypothesis generation* with *controlled experiments*, enabling a predictive and theory-grounded understanding beyond empirical trial-and-error.
> 3. We extend classical models to LLM settings, incorporating finite-size effects ($N$) and heterogeneous interactions across architectures, which are typically not considered in traditional social dynamics models.
>
>
> ---
>
>
> ### Question (and Weakness) 3
> > How should a practitioner use this framework?
>
> Thank you for the opportunity to clarify this point. By identifying the critical temperature $T$ at which biased consensus emerges, our model enables predictive control of multi-agent LLM. Once the model parameters are estimated from a single multi-agent run, we can predict the critical regime under new configurations, including heterogeneous agent mixtures and interaction sparsity. This allows practitioners to move beyond trial-and-error and anticipate when biased consensus will emerge. Most importantly, it can also identify *when no parameter regime is likely to be safe, in which case a multi-agent setup itself may be unsuitable.*
>
>
> ---
>
>
> ### Question 4
> > Is bias inversely related to performance?
>
> This relationship is not universal and mainly arises in complex, open-ended tasks where diverse perspectives are needed (e.g., policy, legal, and clinical decision-making). In such settings, collective bias reflects a collapse of cognitive diversity, where agents converge due to interaction rather than deliberation. In contrast, for simple deterministic tasks with a single ground truth, convergence is not inherently harmful, and this relationship may be weaker or not hold.
>
> ---
>
> We appreciate your openness to reconsidering the score and hope our responses (addressing all concerns and adding new empirical evidence) support a higher evaluation.

---

> > ### Author Rebuttal · Reviewer_bYP4 · 2026-04-03
> >
> > I thank the authors for addressing most of my concerns. I am not fully convinced by the bias argument but trying to address this would require a significant amount of work. I have updated my score to one that I think is appropriate.

---

### Official Review · Reviewer_W7Ks · 2026-03-12

**Soundness:** 3
**Presentation:** 3
**Significance:** 2
**Originality:** 3
**Overall Recommendation:** 4
**Confidence:** 2

**Summary:**

The paper studies how interactions between LLM agents in debate settings can amplify individual biases into collective biased consensus. It proposes a statistical-physics-inspired model of debate dynamics and validates its predictions through controlled experiments with multiple LLMs.

**Compliance With Llm Reviewing Policy:**

Affirmed.

**Final Justification:**

Rebuttal addressed my main concerns.

**Key Questions For Authors:**

The authors may address the weaknesses listed above.

**Limitations:**

Yes.

**Strengths And Weaknesses:**

**Strengths**

S1. Interesting and timely problem.

S2. Simple and interpretable theoretical framework inspired by statistical physics to analyze these dynamics.

S3. The theoretical predictions are supported by controlled experiments across multiple LLM families and tasks.

S4. The paper is generally clear and well written, with experiments designed to illustrate the key mechanisms discussed in the theory.


**Weaknesses:**

Modeling limitations

W1. The paper models debate as a discrete choice process, ignoring the internal autoregressive language generation of LLMs. While this simplification enables theoretical analysis, it may not fully capture real LLM interaction dynamics.

W2. The mean-field model assumes simple interaction networks and a scalar bias field. This may not capture richer communication structures (e.g., roles or asynchronous interaction) or multi-dimensional biases present in real multi-agent LLM debates.

Experiments

W3 (Minor). The investment experiment is not a full backtest. Returns are computed over a single short evaluation window (~5 weeks) without a time-consistent trading simulation or clearly defined asset universe, making the performance results difficult to interpret.

W4. Several experimental claims rely on averages over ~20–50 runs and report only mean trajectories with SEM. For stochastic consensus dynamics, it would be useful to analyze consensus-time distributions (how many rounds runs take to reach agreement), trajectory variability across runs, and finite-size scaling with the number of agents N.

W5. The inferred parameters (e.g., interaction strength \lambda and bias \gamma) may be sensitive to prompt design and decoding settings. For example, prompts encouraging sycophancy or different decoding choices (temperature vs. top-p) could change conformity behavior and affect the estimated parameters.

---

> ### Author Rebuttal · Authors · 2026-03-30
>
> We thank the reviewer for the insightful comments and address each point below.
>
> ---
> ### W1. Discrete choice vs. autoregressive generation
> We agree that modeling the internal generation process is a compelling direction for future research. However, as a foundational first step, our study focuses on the manifested behavior of individual LLMs, specifically their final decisions in discrete choice tasks, and how these behaviors aggregate into macro-level system dynamics. The fact that our simplified model successfully predicts the transition behavior observed in experiments using actual LLMs suggests that this abstraction captures the essential dynamics of the multi-agent system.
> Looking ahead, we plan to generalize our framework to continuous-valued spins (e.g., $O(n)$ models) to capture the internal token-by-token generation. By treating each decoding step’s logit distribution as a state variable, we can model how social influence acts as a time-dependent bias that interferes with the autoregressive transition $P(w_t | w_{<t}, T)$ in real-time. We would very welcome any suggestions from the reviewer on how to further refine this bridge between collective dynamics and internal generation.
>
> ---
> ### W2. Simplified interaction structure
> We agree that real debates involve richer structures (roles, asynchrony, multi-dimensional biases). Our current model serves as a minimal baseline to isolate the dominant factors: social conformity and temperature.
> The framework is extensible: roles can be modeled via asymmetric interactions $J_{ij}$,  agent heterogeneity via additional parameters, and multi-dimensional biases via vector-valued fields (e.g., $O(n)$ models [Friedli et al., 2017]). We will clarify these potential extensions in the revision.
>
> > [Friedli et al., 2017] Friedli, S. and Velenik, Y. Statistical mechanics of lattice systems: a concrete mathematical introduction. Cambridge University Press, 2017
>
> ---
> ### W3. Investment experiment
>
> Thank you for this insightful comment. We agree this is not a full backtest. While agents were prompted to ‘output valid tickers retrievable via Yahoo Finance’, the asset universe is not formally defined. The goal is not financial performance evaluation but a proof-of-concept showing how multi-agent dynamics induce bias in market-like settings. We will explicitly reframe this experiment as preliminary.
>
> ---
> ### W4. Limited statistical analysis
> We thank the reviewer for this valuable suggestion. Following this comment, we extended our analysis to include consensus-time distributions, which characterize stochastic consensus dynamics beyond mean trajectories. As reported in Table 3 below, we find that consensus is reached very early across tasks: the median consensus time is 1-2 rounds, with interquartile ranges typically within 2-3 rounds, and 93-100% of runs reaching agreement by round 3. These results indicate both rapid convergence and low variability across runs.
>
> **Table 3: Effect of Interaction Rounds**
>
> | Task | Median | Mean | Q25 | Q75 | Fraction <= 3 rounds |
> |---|---:|---:|---:|---:|---:|
> | Binary Choice | 1 | 1.0 | 1 | 1 | 100% |
> | Implicit Bias | 2-3 | 2.0-3.0 | 2.0 | 2.0-3.5 | 50%-100% |
> | LLM-as-a-judge | 1 | 1.01-1.37 | 1 | 2 | 93%-100% |
> | Investment Rec. | 2 | 2.0-2.68 | 2.0 | 2.0-3.75 | 99%-100% |
>
> We agree that finite-size scaling w.r.t. $N$ is important. Due to compute limits, we defer this to the camera-ready version.
>
> ---
>
> ### W5. Sensitivity to prompts and decoding
> Thank you for this important point. Rather than assuming invariance to prompts or decoding, our framework is designed to *absorb these variations into a unified set of effective parameters*. Specifically, prompt-induced effects such as sycophancy or instruction-following behavior are reflected in the conformity parameter $\lambda$, while decoding choices (e.g., temperature, top-p) are captured by the noise parameter $\beta$. Thus, our contribution is not that $\lambda$ and $\beta$ are universal constants, but that, within a given configuration, they provide a predictive and interpretable representation of collective dynamics. This allows different prompting and decoding strategies to be compared within a common framework.
>
> ---
>
> We hope that the new experiments and analysis, together with the clarifications above, help address your concerns.

---

> > ### Author Rebuttal · Reviewer_W7Ks · 2026-04-04
> >
> > Thank you. Updating my score.

---

### Official Review · Reviewer_2UJw · 2026-03-13

**Soundness:** 3
**Presentation:** 3
**Significance:** 3
**Originality:** 3
**Overall Recommendation:** 4
**Confidence:** 3

**Summary:**

This paper studies how bias can emerge and amplify in multi-agent large language model (LLM) debate systems. Multi-agent debate is often proposed as a way to improve reasoning by letting several models deliberate before producing a final answer. The authors argue that the interaction itself can introduce new issues. In particular, even when individual models only show modest bias, the debate process can push the group toward a shared biased decision.

The paper illustrates this empirically using two settings: investment recommendation and LLM-as-a-judge evaluation. In the experiments, agents interact over several rounds and gradually move toward a shared decision.

At low sampling temperature the agents converge very quickly. Often within one or two rounds. And once they converge they tend to stay there.

With higher temperature there is more randomness in the outputs and the system seems less likely to lock in early.

To explain this behavior the authors introduce a theoretical framework inspired by statistical physics models of social dynamics. Each agent is modeled as holding a discrete decision state and updates depend on both intrinsic bias and influence from other agents. The model predicts something like phase transition behavior: when conformity becomes large relative to noise the system moves into a state where most agents align.

The rest of the paper attempts to validate this idea through controlled experiments and a few decision-making tasks.

**Compliance With Llm Reviewing Policy:**

Affirmed.

**Final Justification:**

Given the substance of the authors' rebuttal, I have improved their score a notch from the original review I posted.

**Key Questions For Authors:**

- The model assumes discrete decision states. How well does this abstraction capture richer outputs such as long-form reasoning or structured arguments?

- How sensitive are the results to different debate protocols (e.g., asynchronous updates or weighted voting)?

- The paper suggests heterogeneous agents reduce biased consensus. Does the effect depend more on architecture differences or training data differences?

- Could prompt design significantly alter these dynamics?

**Limitations:**

The paper acknowledges several limitations. The theoretical model simplifies LLM behavior by discretizing outputs and assuming simple interaction dynamics.

Empirical evaluation also focuses on small number of tasks and debate protocols.

Still unclear how these dynamics scale to larger agent populations or more realistic deployments.

**Strengths And Weaknesses:**

### Strengths

- The paper raises a useful point. Interaction between multiple LLMs does not necessarily cancel bias and may actually amplify it. Given how often debate-style prompting is proposed as a robustness technique this is worth studying. At the very least it highlights that multi-agent systems can introduce their own dynamics. Not always obvious.


- The connection to spin models from statistical physics is interesting. The analogy is simplified but it gives a framework for thinking about consensus formation. I also appreciate that the paper tries to connect empirical behavior with a mechanistic explanation.

- The work does not rely only on empirical observations. A model is proposed and then tested through synthetic experiments as well as some more realistic tasks. It's good to see both sides.

- A few system design insights come out of the analysis. For example increasing temperature, introducing heterogeneous agents, reducing interaction density. Simple knobs. That is nice.

---

### Weaknesses

- The central empirical result is that biased consensus appears when sampling temperature is low. In Fig. 1 the agents converge to the same answer very quickly when T drops below roughly 0.5. The experiments show this clearly. But it is also somewhat expected. Low temperature removes randomness from sampling, so agents repeatedly generating similar outputs will naturally align. Not clear the phenomenon itself is very surprising. The theory helps frame it, but the core observation still feels somewhat obvious.

- The proposed model represents responses as discrete states in a spin system. This makes the analysis tractable but removes a lot of what happens in real LLM debates. Actual debate prompts involve long text outputs, reasoning chains, structured prompts, etc. Influence between agents is probably not symmetric either. So there is a bit of gap here. The model captures a stylized version of the dynamics. Not necessarily wrong, just simplified.

- The empirical validation focuses mainly on two tasks: investment recommendation and LLM-as-a-judge evaluation. Interesting examples, but still fairly narrow. It would help to see if similar dynamics appear in more complex reasoning tasks where debate systems are typically used. Right now it is a bit hard to judge generality.

- In the investment experiment bias is measured as the proportion of US / technology stocks in the generated portfolio. Not completely obvious this represents undesirable bias rather than patterns in the underlying market or training data. Worth clarifying. A short discussion here would help.

- The experiments appear to use relatively small populations (e.g., six agents in the MT-Bench judge experiments) but theoretical discussion relies on mean-field style formulation. Would be helpful to see whether the observed dynamics remain stable when number of agents changes. Right now the link between theory and experiment is a little thin.

- Most experiments run only small number of rounds (around seven). System appears to lock into consensus very early anyway. Still, would be interesting to see whether longer interactions change the behavior. Or maybe just reinforce it.

---

> ### Author Rebuttal · Authors · 2026-03-30
>
> We thank the reviewer for the insightful comments.
>
> ---
> ### Weakness 1
> While convergence itself may be expected, our key finding is that even *arbitrarily small initial biases are non-linearly amplified*, leading to bias lock-in. Our contribution goes beyond observing convergence: we identify when it occurs (critical regime) and how it scales. Importantly, the system exhibits a sharp crossover regime, which is not obvious a priori. This has direct implications for the safety of multi-agent systems.
>
> ---
> ### Weakness 2 & Question 1
> Many real-world decision-making tasks (e.g., ranking, selection, voting) reduce to discrete choices, even when intermediate reasoning is complex, which our theory captures. Moreover, controlled experiments with real LLMs reproduce key phenomena predicted by our model. We therefore view this as a necessary minimal model that isolates the core mechanism. Future work will generalize to continuous-valued spins (e.g., $O(n)$ models), using logits as $\sigma_i$, and incorporating asymmetric interaction structures.
>
> ---
> ### Weakness 3
> We would like to clarify that our study evaluates four tasks (rather than two): investment, LLM-as-a-judge, binary token choice, and gender bias. Across all four tasks, we observe consistent behavior, supporting the generality of our findings.
>
> Inspired by the reviewer's comment, we conducted additional experiments on OpinionQA (https://github.com/tatsu-lab/opinions_qa) in a multi-agent debate setting. Here bias is clearly defined here as favoring the ideologies of majority demographic groups over those of minority groups.
>
> **Table 1: Final Bias across Temperatures (T)**
> | Category | 0.0 | 0.1 | 0.5 | 1.2 |
> |---|---:|---:|---:|---:|
> | Party | 0.451 | 0.388 | 0.335 | 0.382 |
> | Race | 0.444 | 0.410 | 0.399 | 0.381 |
> | Education | 0.436 | 0.417 | 0.417 | 0.379 |
> | Gender | 0.655 | 0.532 | 0.571 | 0.496 |
>
> Bias is highest at T=0.0. Increasing T (>0.1) reduces bias.
>
> **Table 2: Bias Evolution**
> (a) T=0.0
> | Category | R1 | R3 | R5 |
> |---|---:|---:|---:|
> | Party | 0.391 | 0.451 | 0.451 |
> | Race | 0.373 | 0.457 | 0.457 |
> | Education | 0.365 | 0.436 | 0.436 |
> | Gender | 0.620 | 0.655 | 0.655 |
>
> (b) T=0.8
> | Category | R1 | R3 | R5 |
> |---|---:|---:|---:|
> | Party | 0.363 | 0.341 | 0.363 |
> | Race | 0.411 | 0.396 | 0.413 |
> | Education | 0.390 | 0.400 | 0.412 |
> | Gender | 0.542 | 0.558 | 0.554 |
>
> At low temperature, bias increases over rounds, indicating bias amplification, while at higher temperature it remains stable. These results are consistent with the main paper and support generality. We will further extend our evaluation to additional high-stakes domains, including hiring and legal decision-making.
>
> ---
> ### Weakness 4
> We agree that concentration could partly reflect underlying market structure or training data. However, this alone cannot explain our observations. The concentration is not static: it increases over interaction rounds only at low temperatures (Fig. 1(a)). This suggests that the extreme concentration (e.g., up to 0.97 in U.S. stocks) arises from interaction-driven amplification rather than solely from underlying data.
>
> ---
> ### Weakness 5
> Our theoretical framework explicitly incorporates finite-size effects, as discussed in Section 5 (“Collective Dynamics”). We model finite-size noise scaling as $\mathcal{O}(1/\sqrt{\rho N})$. As a result, the sharp phase transition in the $N \to \infty$ limit is naturally smoothed into a crossover regime for finite $N$, consistent with our empirical observations.
>
> ---
> ### Weakness 6
> Most dynamics occur within the first few rounds (see **Table 3** in our response to Reviewer W7Ks). Post-consensus states are highly stable: 99.25% (LLM-as-a-Judge) and 97.75% (investment) of runs show negligible deviation (<0.1), with similar stability in synthetic settings (100.0% for Binary Choice, 98.75% for Implicit Bias). These results indicate that extending the number of rounds does not qualitatively change the outcomes.
>
> ---
> ### Question 2
> Fig. 4(a) shows that modifying interaction structure (e.g., sparsity) shifts behavior via effective interaction strength. Asynchronous interactions can be incorporated through the interaction matrix $J_{ij}$, and weighted voting by extending this term with weights, without altering the qualitative behavior.
>
> ---
> ### Question 3
> Our framework operates at the behavioral level, where heterogeneity is captured by parameters such as conformity and intrinsic bias. Fig. 5 shows that conformity varies across model families, suggesting an architectural contribution, while intrinsic bias varies more within families, indicating additional effects from training and alignment.
>
> ---
> ### Question 4
> Prompting modulates dynamics via effective conformity $\lambda$: as shown in Fig. 4(d), increased sycophancy induces biased consensus, while weaker conformity mitigates it.
>
> ---
>
> We hope that the new experimental results and clarifications help address your concerns.

---

> > ### Author Rebuttal · Reviewer_2UJw · 2026-04-02
> >
> > I appreciate the authors’ detailed responses and the additional experiments in the rebuttal. The clarification around the helps better frame the significance of the results. The added experiments also strengthen the empirical evidence and address my concerns about generality.
> >
> > Overall, the rebuttal resolves most of my concerns and improves my understanding of the contribution.
> >
> > I plan to adjust my score accordingly.

---

### Decision · Program_Chairs · 2026-04-30

**Decision:**

Accept (regular)

**Comment:**

This paper studies how interactions in multi-agent LLM debates can produce collective biased consensus. It introduces a statistical-physics-inspired analytical framework in which agents’ decisions evolve under intrinsic bias, social conformity, and sampling noise, and uses this framework to characterize when debate dynamics transition from weakly biased behavior to stable collective lock-in. The paper evaluates these predictions through controlled experiments and additional decision-making settings, including investment recommendation and LLM-as-a-judge, and also studies how factors such as temperature, interaction structure, and agent heterogeneity affect the emergence of biased consensus.

The reviewers were broadly positive and viewed the paper as timely, clearly written, and strengthened by the combination of an interpretable theoretical framework with controlled empirical validation. The main concerns were about the simplifying assumptions of the model, the scope and generality of the empirical settings, the interpretation of bias in some realistic tasks, and the accessibility of the theory to readers without a statistical-physics background. In the rebuttal and discussion, the authors provided additional experiments and analyses, clarified the intended role of the realistic tasks and the generality of the observed dynamics, and gave a concrete plan for revising the presentation of the theory to make it more pedagogical. Overall, the discussion remained clearly on the positive side, and I recommend acceptance.